# Factors associated with stroke recurrence and mortality in ischemic and haemorrhagic stroke: Protocol for a systematic review and meta-analysis

Simone Ryan[1,2]*, Katie Robinson[1,2,3], Rose Galvin[1,2,3], Margaret O'Connor[2,4,5], John McCabe[6,7], Eleanor Fallon[2], Sara Hayes[1,2,3]

1 School of Allied Health, Faculty of Education and Health Sciences, University of Limerick, Limerick, Ireland, 2 Health Research Institute (HRI), University of Limerick, Limerick, Ireland, 3 Ageing Research Centre (ARC), University of Limerick, Limerick, Ireland, 4 Department of Ageing and Therapeutics, University of Limerick Hospital Group, Limerick, Ireland, 5 School of Medicine, Faculty of Education and Health Sciences, University of Limerick, Limerick, Ireland, 6 School of Medicine, University College Dublin, Dublin, Ireland, 7 Stroke Service, Department of Geriatric Medicine, Mater Misericordiae University Hospital, Dublin, Ireland

* ryan.simone@ul.ie

## Abstract

### Background and objectives

Stroke is one of the leading causes of death and disability worldwide. People with stroke face a significant risk of recurrence, however, there is a lack of consensus on the risk factors for stroke recurrence. Previous systematic reviews in this area are either outdated, focus exclusively on clinical risk factors, or are limited by geographical region, stroke subtype or age group. This review aims to systematically identify clinical and lifestyle risk factors associated with stroke recurrence and mortality in people with stroke.

### Methods

A systematic search will be conducted in four electronic databases (PubMed, EMBASE, CINAHL and PsycINFO) and grey literature sources. Only prospective cohort studies that estimate the association between risk factors and stroke recurrence and/or mortality in adults with ischemic and haemorrhagic stroke will be included. The exposures of interest are clinical and lifestyle risk factors, including vascular and cardiac factors such as hypertension or atrial fibrillation, and behavioural factors such as physical activity, nutrition, or smoking. Two independent reviewers will conduct screening, data extraction, and risk of bias assessment using the ROBINS-E tool, while the GRADE approach will be used to evaluate the certainty of evidence. A meta-analysis using a random-effects model will be used to determine the overall effect for each exposure on stroke recurrence and/

**Data availability statement:** No datasets were generated or analysed during the current study. All relevant data from this study will be made available upon study completion.

**Funding:** SR is a PhD scholar funded by the Health Research Board SPHeRE/2022/1. The funding body was not involved in the design of this protocol, the decision to publish, or the preparation of this manuscript. Funder website: https://www.hrb.ie/grant-approved/structured-population-and-health-services-re-search-education-sphere-phase-iii/

**Competing interests:** The authors have declared that no competing interests exist.

or mortality. However, if a meta-analysis is not justified, a narrative synthesis will instead be conducted.

## Discussion

Recurrent strokes are associated with greater mortality and poorer functional prognosis than first-time events, imposing greater economic burden on healthcare services. This review will identify factors that increase one's risk for recurrence, thereby informing the development and refinement of clinical guidance for secondary stroke prevention; a key priority for both those living with stroke and the wider health service.

## Introduction

Stroke is a leading cause of death and disability worldwide [1] accounting for 11.6% of all deaths in 2019 [2]. Globally, over 12.2 million new strokes occur annually [1], and survivors face a significant risk of recurrence, with recurrence rates estimated at 11.1% and 39.7% at 1 and 12 years respectively [3].

In recent decades, substantial progress has been made in the prevention, diagnosis and treatment of cardiovascular disease and stroke, leading to significant reductions in the incidence of first and recurrent stroke [4–7]. However, the rising prevalence of known vascular risk factors such as diabetes mellitus, hypertension, and unhealthy lifestyles [8,9], combined with suboptimal and fragmented prevention strategies [10–12], threatens to undermine these advances.

Recurrent stroke is associated with greater mortality and poorer functional prognosis than first-time events, as evidenced by Skajaa et al.'s [13] prospective, population-based cohort study (n = 105,397). A recent systematic review (n = 9) of the costs of recurrent stroke also demonstrated recurrence imposes greater economic burden on healthcare services [14]. Therefore, prevention of recurrence is a key priority for both survivors [15] and the wider health service.

While risk factors for first-time stroke are well established [16], there is a lack of consensus regarding the risk factors for stroke recurrence. This discordance is reflected through inconsistencies across international clinical guidelines for secondary prevention of stroke [17–20]. Several systematic reviews have established associations between known vascular risk factors, such as atrial fibrillation and diabetes mellitus and stroke recurrence [21–24]. However, these reviews exhibit several limitations.

Firstly, Zheng & Yao's [21] systematic review and meta-analysis (n = 11) established associations between stroke recurrence and hypertension, diabetes mellitus, atrial fibrillation, and coronary heart disease; however, this review focused exclusively on clinical risk factors, to the neglect of lifestyle risk factors such as nutrition, physical activity, and other emerging risk factors such as migraine [17–20]. Furthermore, this review is restricted in scope to risk factors associated with ischemic stroke recurrence, therefore limiting the generalizability of results to haemorrhagic stroke. While lifestyle factors such as smoking and alcohol consumption are addressed by other

systematic reviews, such as those by Nindrea & Hasnuddin [22] (n = 11) and Kolmos et al. [23] (n = 26), both reviews again focus solely on ischemic stroke, and are limited in the range of lifestyle factors evaluated. Furthermore, identification of risk factors for recurrence was not the primary focus of Kolmos et al. [23] (n = 26), potentially restricting the depth and breadth of findings.

Previous reviews in this area are also restricted in scope by age or geographical location; Chiangkong et al.'s [24] systematic review and meta-analysis (n = 17) focuses on risk factors for recurrence in the working-age population, with Mbalinda et al. [25] limited to risk factors for recurrence in sub-Saharan Africa. Additionally, some of these reviews are outdated, indicating the need for an updated overview of the literature [21,23].

While multiple systematic reviews have evidently been conducted on the topic of risk factors associated with stroke recurrence, it is not deemed appropriate to complete an umbrella review, as several factors identified across international clinical guidelines to be a priority for secondary prevention are not evaluated by these reviews. These factors include several lifestyle factors such as physical activity, nutrition, and substance use, as well as various clinical factors, including, but not limited to migraine and sleep apnoea. Therefore, an umbrella review of existing reviews would be restricted in scope. This protocol will therefore describe a systematic review which will address these issues by producing an up-to-date synthesis and meta-analysis of the existing evidence. The proposed review will establish the association between a comprehensive set of risk factors and stroke recurrence and/or mortality in both ischemic and haemorrhagic stroke populations.

### Review aim

This systematic review and meta-analysis will aim to identify the clinical and lifestyle risk factors associated with stroke recurrence and mortality in adults who have experienced either ischemic or haemorrhagic stroke.

### Review questions

a) What are the factors associated with stroke recurrence in adults with ischemic and haemorrhagic stroke?

b) What are the factors associated with mortality following stroke recurrence in adults with ischemic and haemorrhagic stroke?

## Methods

The protocol for this systematic review and meta-analysis was registered prospectively in PROSPERO (CRD42024610123) in December 2024. The development of this protocol has been based on standards of the Cochrane Collaboration [26] and the Preferred Reporting Items for Systematic Reviews and Meta-Analyses Protocols (PRISMA-P) guidelines [27]. (S1 Table).

### Eligibility criteria

Inclusion and exclusion criteria for this review will be detailed below in line with the Population-Exposure-Comparator-Outcome-Study Design (PECOS) framework [26,28].

**Population.** This review will consider studies of individuals with a formal diagnosis of stroke. Stroke is defined according to the World Health Organization criteria, as 'rapidly developed clinical signs of focal (or global) disturbance of cerebral function, lasting more than 24 hours or leading to death, with no apparent cause other than of vascular origin' [29]. Stroke subtypes eligible for review will include ischemic strokes (such as large artery thrombosis, cardiogenic embolism and cryptogenic stroke), transient ischemic attacks (TIA), and intracerebral haemorrhage (ICH). Subarachnoid haemorrhage (SAH) will be excluded as the majority of SAH are secondary to aneurysm rupture, thus differing in aetiology to ICH and potentially introducing heterogeneity into analysis.

Only adults aged 18 years or older will be included. Individuals under the age of 18 will be excluded. For studies that include participants from both age groups, inclusion will be considered if data for participants aged 18 or older are reported separately. Where data are instead aggregated, contact will be made with the study authors for separated data. If separated data are not provided, the study will be excluded. Individuals without a formal diagnosis of stroke will also be excluded.

**Exposure.** The exposures of interest for this review are clinical and lifestyle risk factors. These are defined as physiological or behavioural factors that may increase one's risk of stroke recurrence. Clinical risk factors include i) vascular risk factors (hypertension; dyslipidaemia; glucose control; obesity; obstructive sleep apnoea; migraine; cerebral amyloid angiopathy; chronic kidney disease) and ii) cardiac factors (atrial fibrillation; patent foramen ovale; myocardial infarction; heart disease). Lifestyle risk factors include i) nutrition, ii) physical activity, iii) smoking, iv) alcohol consumption, v) substance use, vi) medication adherence (statin and antiplatelet therapy).

These exposures were chosen as eligible for inclusion in this review following a series of consultations with relevant stakeholders and existing literature. Firstly, international clinical guidelines addressing secondary prevention of both ischemic and haemorrhagic stroke were reviewed for frequently occurring clinical and lifestyle risk factors; exposures referenced across at least two guidelines were prioritized for inclusion [17–20]. Further relevant risk factors were identified through review of existing population-based studies of ischemic and haemorrhagic stroke recurrence [13,30–32], alongside evaluation of previous systematic reviews for gaps where prioritized risk factors had not been addressed [21–25]. Potential exposures of interest were also identified through consultation with Public Patient Involvement (PPI) panel members. Expert stakeholders were then consulted to reach a consensus on the chosen exposures for review.

Demographic factors such as age, gender or stroke severity will not be considered as exposures in this review; instead, further analyses will be conducted to compare recurrence risk across demographic groups, including stroke subtypes, which will be discussed further in later sections of this protocol. Neuroimaging predictors and biomarkers of stroke recurrence will be excluded as exposures as they are outside the scope of the present review.

**Comparator.** This review will not place inclusion or exclusion criteria on a comparator variable.

**Outcome.** The primary outcomes of this review are recurrent stroke and/or mortality due to stroke recurrence. Stroke recurrence will be classified according to the same criteria as the initial stroke and is further defined by the onset of new and persisting (>24 hours) neurological deficits after the index stroke that are not associated with other systemic or neurological causes [29]. New events in the same vascular territory as the first event are considered recurrent when clearly associated with new neurological symptoms. Progressive or worsening stroke and neurological deterioration caused by other factors such as haemorrhagic transformation, brain oedema, seizure, pneumonia, sepsis, heart or respiratory failure are not considered as stroke recurrence. Diagnosis of stroke recurrence should be certified by a relevant medical professional based on history, clinical presentation, and/or findings in neuroimaging. Stroke mortality is defined as deaths which occurred due to stroke after 24 hours of hospital admission [33]. Where data are available, stroke recurrence outcomes will be presented separately according to subtype of recurrence, i.e., haemorrhagic stroke after incident ischemic stroke, or haemorrhagic stroke after incident haemorrhagic stroke etc.

**Study design.** Prospective cohort study designs reporting data of an etiological nature will be considered for inclusion in this review. This will include population-based studies that employ a prospective cohort design. Other observational study designs such as retrospective cohort studies and/or case-control studies will be excluded, as their retrospective nature may introduce various biases that could compromise confidence in the review's findings. Cross-sectional study designs will also be excluded, as they cannot establish etiological associations. Interventional designs such as randomised controlled trials or quasi-experiments will not be included, as while they may present data of an etiological nature, the outcome may be confounded by the intervention, and therefore conclusions regarding associations between exposure and outcome may be limited.

**Study Context.** No restrictions will be placed on the study context for this review. All contexts will be considered for inclusion, i.e., country of origin, hospital or community settings.



## Search strategy

A comprehensive search strategy will be employed, with both published and unpublished literature considered for inclusion in this review. Four electronic databases will be searched (Pubmed, Embase (OVID), CINAHL (EBSCO) and APA PsychINFO). The following search terms will be used: (stroke OR ischemic stroke OR embolic stroke OR thrombotic stroke OR haemorrhagic stroke OR transient ischemic attack OR cerebrovascular accident) AND (mortality) AND (recurrence OR relapse) AND (risk factor OR risk). S2 Table details a draft form of the PubMed strategy.

Trial registration databases such as clinicaltrials.gov and WHO ICTRP will also be searched for unpublished/ongoing observational studies that are eligible for inclusion, with contact made to authors to obtain relevant data where possible. The reference lists of relevant studies and reviews will be hand searched. Citation searching of relevant studies will be completed through Google Scholar and/or Scopus. No restrictions will be placed on publication language; the full text of non-English studies retrieved will be translated to English using freely available, high-quality translation technologies, such as DeepL Translate, and then screened for eligibility. No restrictions will be placed on the search in terms of publication date. The search will be completed by April 2025.

## Study selection

Following the search, results from each source will be uploaded to an online systematic review management software, where duplicates will be removed. Titles and abstracts of all reports will then be screened in line with eligibility criteria, with the full text of potentially relevant reports retrieved and examined in line with eligibility criteria. Reasons for exclusion of studies at this stage will be documented and recorded. Reviewers will correspond with study authors where appropriate to clarify study eligibility and request additional information. The screening process will be completed by two independent reviewers. Disagreements regarding study eligibility will be resolved through either discussion or referral to a third member on the review team. Results of the selection process will be depicted in a PRISMA flow diagram. The screening process will be performed through Covidence.

## Data extraction

Data will then be extracted from the final set of included studies. Data extraction will be completed independently by two reviewers and will be performed through Covidence. Covidence will be used to create a data extraction template. This template will be piloted, and any disagreements will be resolved via discussion or referral to a third review team member. The following details will be extracted from each study: study details (author, year, journal); participants (sample size, participant demographics, i.e., age, gender, ethnicity, socioeconomic status, clinical history); study method/characteristics (study design, setting, recruitment procedures utilized, follow-up or study duration, exposures of interest); outcome (primary and secondary outcomes relevant to review objectives); data analysis methods; adjustment for confounding factors (e.g., age, gender, stroke severity, co-morbidities); study results (appropriate measures for effect size such as adjusted and unadjusted risk ratios, relative ratios, hazard ratios, odds ratios, p-values, confidence intervals); other relevant study information (e.g., funding sources, conflict of interest). Where data are ambiguous or missing, contact will be made with study authors where possible.

## Risk of bias assessment

The ROBINS-E tool will be used to assess for bias due to confounding, bias arising from measurement of the exposure, bias in selection of participants into the study, bias due to post-exposure interventions, bias due to missing data, bias arising from the measurement of the outcome, and bias in selection of the reported results.

Meta-biases will also be assessed in this review. A funnel plot will be used to visually explore potential publication bias. In the presence of asymmetry, reviewers will investigate possible explanatory factors including possible missing studies and study quality.

## Data synthesis and analysis

Data extracted via Covidence will be exported to RevMan 5 where data synthesis will take place. Firstly, the characteristics of each study will be summarized in a 'characteristics of included studies' table. Similar studies will be grouped within each exposure to compare characteristics and effects across studies.

Primary analysis will consider all strokes, however subgroup analyses will be completed to compare the risk of recurrence and mortality within certain clinical and demographic subgroups, such as ischemic stroke and haemorrhagic stroke; working age (18–64 years) and older adults (over 65 years); male and female participants; and participants of low, middle and high socioeconomic status (where data are available). Where appropriate, meta-regression will be used in place of subgroup analysis to analyse the effect of continuous study-level characteristics, such as year of publication or participant characteristics, on review outcomes.

If the studies included are sufficiently homogenous with regards to clinical and methodological factors, a meta-analysis will be conducted to determine the overall effect of each exposure on the review outcomes. Study outcomes of stroke recurrence and mortality are measured as dichotomous outcomes. Odds ratios and hazard ratios are deemed the most appropriate effect measures for prognostic studies with rare outcomes and will be combined for analysis as they approximate a similar level of risk [34]. This will be completed as it is anticipated that there may be variation in the effect measures used across included studies. Furthermore, where multiple adjusted measures are reported in a study, the most adjusted effect measure will be used for analysis. Variables included in the adjustment will be documented. It is also possible that, due to variation in reporting of effect measures across studies, crude effect estimates may need to be calculated manually prior to analysis.

Separate meta-analyses will be conducted for each exposure of interest, with studies grouped within each exposure. The random-effects model is chosen for meta-analysis, as the fixed-effects model ignores the possibility of statistical heterogeneity, and results of the random-effects model are almost identical to that of the fixed-effects model when there is no heterogeneity present. Therefore, as it is anticipated that the range of studies included will introduce heterogeneity into analysis, a random-effects model is preferable. During meta-analysis, the random-effects model will be used to generate the effect estimate with 95% confidence interval to evaluate the associations between each exposure (risk factor) and recurrent stroke/mortality. Results of each meta-analysis will be presented in a forest plot. The level of statistical heterogeneity will be assessed using the $I^2$ statistic. If, however, the level of heterogeneity exceeds 75%, a meta-analysis will not be appropriate and instead a narrative synthesis of results will be conducted, including summaries of the effect estimates for each study. Sensitivity analyses will be used to assess the impact of assumptions made throughout analysis, for example, the effects of random vs fixed effects models during analysis, omission of studies based on risk of bias and results obtained from pooling adjusted and unadjusted effect measures. Subgroup analyses and meta-regressions completed will also aid identification of potential sources of heterogeneity. It is not possible to predict all sensitivity and subgroup analyses that may need to be undertaken for this review; issues suitable for sensitivity analysis may arise during the review process in relation to the individual studies under investigation. Sensitivity and subgroup analyses undertaken that were not pre-specified in the protocol will be clearly reported with rationale in the final review.

## Assessment of confidence in findings

The GRADE approach (Grading of Recommendations, Assessment, Development and Evaluation), as recommended by the Cochrane Collaboration [35], will be used to assess the certainty of evidence for this review. The GRADE approach specifies four levels of certainty for a body of evidence for a given outcome: high, moderate, low and very low. These assessments are determined through consideration of five domains: risk of bias, inconsistency, indirectness, imprecision and publication bias. As this review will include observational studies only, these studies will start with a low quality-of-evidence level, as established by the tool [36]. However, three further criteria established by the GRADE system will also be considered which can raise the level of evidence for observational studies: large magnitude of effect, residual effect of



confounding variables and dose-response gradient. GRADE assessments will be completed for each synthesis of evidence in this review, i.e., each meta-analysis and/or narrative synthesis.

A summary of findings table will be formulated to present the findings of the review in a transparent and structured format. This table will provide key information regarding the strength of association between each exposure and outcome, as well as the amount and certainty of evidence available.

### Ethical considerations and plans for dissemination

Ethical approval is not required for this study as it involves the analysis of publicly available data with no personally identifiable information. It is intended to submit this review for publication in a peer-reviewed journal following completion. The results of this review will be disseminated through conference presentations, dissemination via organizational newsletters, and dissemination to relevant stakeholder groups.

### Public-Patient-Involvement (PPI)

An existing PPI panel of people with stroke and their family members was directly consulted by the lead author on a singular occasion during the planning stage of this review. Potential exposures of interest to be included in the review were identified through this consultation, as well as the identification of potential terms of relevance for the search strategy. Stakeholder level of involvement during this process can be described as 'contributing', as per the ACTIVE framework for reporting stakeholder involvement in systematic reviews [37].

### Status and timeline of review

At the time of submission, this review is yet to commence. This review is anticipated to start March 2025. It is anticipated this this review will be complete and results disseminated within 12 months of the initial commencement date. Any amendments made to the protocol following publication will be clearly documented and justified in the final report.

## Discussion and conclusions

This systematic review and meta-analysis will aim to comprehensively identify the risk factors associated with stroke recurrence and mortality in adults with ischemic and haemorrhagic stroke and transient ischemic attack. The findings of this review will address key gaps and inconsistencies in the literature, including discordance across international clinical guidelines regarding the risk factors for stroke recurrence [17–20]. Additionally, it will expand upon previous systematic reviews, which have predominantly focused on clinical risk factors, such as hypertension or atrial fibrillation [21–25], often neglecting lifestyle factors, such as physical activity and nutrition, and other, emerging risk factors such as migraine or sleep apnoea. Many prior reviews have also been restricted by geographical scope, age, or stroke subtype, and several are now outdated.

By providing a comprehensive and aggregated understanding of the risk factors for recurrence and mortality, this review will support the development of community- and population-based prevention efforts, a cornerstone of both primary and secondary stroke prevention [38,39]. Simultaneously, the planned subgroup analyses will identify distinct risk factors by age, gender, stroke subtype, and socioeconomic status, aiding the development of personalized prevention strategies, which are a key focus of international stroke initiatives such as the European Stroke Action Plan 2018–2030 [40] and national stroke strategies, such as the Irish National Stroke Strategy 2022–2027 [41].

The findings of this review are also expected to inform and refine existing clinical guidelines for secondary stroke prevention, thus reducing variations in practice; improvement in the consistency of care following refinement of these guidelines could significantly improve patient outcomes. Moreover, individuals who experience a stroke often face an elevated risk of subsequent cardiovascular events and vascular death. Since there is overlap across vascular risk factors for stroke and other cardiovascular diseases such as myocardial infarction, the vascular risk factors identified through this review may also help inform broader cardiovascular disease prevention efforts [42].

Given the global burden and significant consequences of stroke recurrence, urgent action is needed to develop effective, evidence-based interventions at both the individual and population levels. This review will thus comprehensively synthesize and pool the existing evidence, providing a foundation for improved and evidence-informed secondary stroke prevention.

## Potential limitations

Some limitations are anticipated in this systematic review. First, studies that are retrospective in nature, or employ a case-control or cross-sectional design, will be excluded. While this may lead to the exclusion of potentially relevant data, their exclusion is justified as including these designs may introduce various biases that could compromise confidence in the review's findings.

Second, the pre-selection of exposures for review may limit the identification of emerging risk factors. However, given the extensive range of potential risk factors for stroke recurrence and mortality in the literature, imposing exclusionary criteria on study exposures was deemed necessary to ensure the review remains focused and feasible within the designated timeframe. The selection of eligible exposures for review was justified through consultations with PPI contributors and expert stakeholders, analyses of secondary stroke prevention guidelines and existing literature, and an assessment of gaps in prior systematic reviews. Despite this focused approach, future research may be required to identify emerging risk factors beyond the scope of this review. Furthermore, important neuroimaging predictors and biomarkers of stroke recurrence will not be included in this review due to scope restraints; future research is thus required to identify the neuroimaging markers associated with stroke recurrence.

Third, due to the wide range of exposures across included studies, variations in effect estimates may arise as a result of differences in how confounders are controlled for in each study. While this will be addressed during data analysis, some variation is unavoidable. Lastly, the exclusion of non-English studies presents a limitation, as potentially relevant studies may be missed. However, this criterion is necessary given the limited translation facilities of the review team.

## Supporting information

**S1 Table. PRISMA-P 2015 checklist.**
(DOCX)

**S2 Table. PubMed Search strategy.**
(DOCX)

## Acknowledgments

Acknowledgements are provided to Liz Dore, health research methods librarian at the University of Limerick, for her guidance with the development of the search strategy for this review. SR and SH were involved in the conception and design of this protocol. SR wrote the original draft of the protocol, which was reviewed and edited by SH, KR, RG, MO'C, JM and EF. EF also contributed to and provided guidance on the data analysis plan. All authors approved the final version for publication. SR is the guarantor of this review.

## Author contributions

**Conceptualization:** Simone Ryan, Sara Hayes.

**Methodology:** Simone Ryan, Katie Robinson, Rose Galvin, Margaret O'Connor, John McCabe, Eleanor Fallon, Sara Hayes.

**Project administration:** Simone Ryan.



**Supervision:** Katie Robinson, Rose Galvin, Sara Hayes.

**Writing – original draft:** Simone Ryan.

**Writing – review & editing:** Simone Ryan, Katie Robinson, Rose Galvin, Margaret O'Connor, John McCabe, Eleanor Fallon, Sara Hayes.

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
