## [Decision Letter · Decision Letter 0]

19 Jul 2025

Dear Dr. Ryan,

Thank you for submitting your manuscript to PLOS ONE. After careful consideration, we feel that it has merit but does not fully meet PLOS ONE’s publication criteria as it currently stands. Therefore, we invite you to submit a revised version of the manuscript that addresses the points raised during the review process.

We look forward to receiving your revised manuscript.

Kind regards,

Eyob Alemayehu Gebreyohannes, PhD

Academic Editor

PLOS ONE

Journal Requirements:

Additional Editor Comments :

Dear Mr Ryan,

Thank you very much for submitting your manuscript, "Factors associated with stroke recurrence and mortality in ischemic and haemorrhagic stroke: protocol for a systematic review and meta-analysis" (Manuscript ID: PONE-D-25-10654), to PLOS One. I also appreciate your patience during what has been an extended review process.

As an Academic Editor, I was unable to secure a second reviewer despite repeated efforts — an outcome that is relatively uncommon but sometimes unavoidable. Nevertheless, after receiving feedback from one expert reviewer and conducting a thorough evaluation by myself, I am confident in moving forward based on the input available.

Following this assessment, I have reached a decision of minor revision. The reviewer’s comments are included below and should guide your revisions. Please provide a point-by-point response to the reviewer’s suggestions, indicating how you have addressed each comment in the revised manuscript.

I look forward to receiving your revised submission.

Kind regards,

Eyob Alemayehu Gebreyohannes (PhD)

Academic Editor

Reviewers' comments:

Reviewer's Responses to Questions

**Comments to the Author**

1. Does the manuscript provide a valid rationale for the proposed study, with clearly identified and justified research questions?

Reviewer #1: Yes

2. Is the protocol technically sound and planned in a manner that will lead to a meaningful outcome and allow testing the stated hypotheses?

Reviewer #1: Yes

3. Is the methodology feasible and described in sufficient detail to allow the work to be replicable?

Reviewer #1: Yes

4. Have the authors described where all data underlying the findings will be made available when the study is complete?

Reviewer #1: Yes

5. Is the manuscript presented in an intelligible fashion and written in standard English?

Reviewer #1: Yes

You may also provide optional suggestions and comments to authors that they might find helpful in planning their study.

Reviewer #1: The authors present a well-designed systematic review protocol aimed at identifying clinical and lifestyle risk factors associated with recurrence of hemorrhagic and ischemic stroke. The justification for the future review is well-founded, and the authors have met all the requirements for a systematic review of observational studies. I congratulate the authors on the quality of the manuscript but suggest that they rethink selecting only articles published in English, since more and more technologies, artificial intelligence, and sites such as DeepL Translate make it possible to translate articles published in other languages into English with good quality. Therefore, in order to avoid this limitation put forward in the discussion, I suggest you review and consider including studies published in any language, with the explanation that they will be translated into English with good-quality free technologies available.

**Do you want your identity to be public for this peer review?** For information about this choice, including consent withdrawal, please see our Privacy Policy

Reviewer #1: **Yes: ** Ricardo Ney Cobucci

---

## [Author Response · Author response to Decision Letter 1]

22 Jul 2025

Dear editor, and review team

Thank you for your time and effort reviewing this article. We appreciate the comments provided by the editor and reviewier #1. We have addressed the journal requirements in relation to style requirements, file naming conventions and supplementary information guidelines. We have also reviewed our reference list and ensured it is correct. We have also addressed the comments provided by reviewer #1 in our manuscript and response letter. Specifically, we have decided to consider non-English studies for inclusion, and to use freely available technologies such as DeepL Translate to translate texts into English for eligibility screening. We hope that these measures are adequate, and we look forward to progressing this submission in your esteemed journal.

Simone Ryan

---

## [Editor Report · Decision Letter 1]

24 Jul 2025

Factors associated with stroke recurrence and mortality in ischemic and haemorrhagic stroke: protocol for a systematic review and meta-analysis

PONE-D-25-10654R1

Dear Dr. Ryan,

We’re pleased to inform you that your manuscript has been judged scientifically suitable for publication and will be formally accepted for publication once it meets all outstanding technical requirements.

Kind regards,

Eyob Alemayehu Gebreyohannes, PhD

Academic Editor

PLOS ONE

---

## [Editor Report · Acceptance letter]

PONE-D-25-10654R1

PLOS ONE

Dear Dr. Ryan,

I'm pleased to inform you that your manuscript has been deemed suitable for publication in PLOS ONE. Congratulations! Your manuscript is now being handed over to our production team.

Kind regards,

on behalf of

Dr. Eyob Alemayehu Gebreyohannes

Academic Editor

PLOS ONE